# Prebiotic Impacts of Soybean Residue (Okara) on Eubiosis/Dysbiosis Condition of the Gut and the Possible Effects on Liver and Kidney Functions

**DOI:** 10.3390/molecules26020326

**Published:** 2021-01-11

**Authors:** Mohammed Sharif Swallah, Hongliang Fan, Sainan Wang, Hansong Yu, Chunhong Piao

**Affiliations:** 1College of Food Science and Engineering, Jilin Agricultural University, Changchun 130118, China; m.s.swallah@gmail.com (M.S.S.); f_hongliang@163.com (H.F.); 18644941109@163.com (S.W.); 2Soybean Research & Development Centre, Division of Soybean Processing, Chinese Agricultural Research System, Changchun 130118, China

**Keywords:** dietary fiber, gut microbiota, kidney, liver, okara, prebiotic, soybean residue

## Abstract

Okara is a white-yellow fibrous residue consisting of the insoluble fraction of the soybean seeds remaining after extraction of the aqueous fraction during the production of tofu and soymilk, and is generally considered a waste product. It is packed with a significant number of proteins, isoflavones, soluble and insoluble fibers, soyasaponins, and other mineral elements, which are all attributed with health merits. With the increasing production of soy beverages, huge quantities of this by-product are produced annually, which poses significant disposal problems and financial issues for producers. Extensive studies have been done on the biological activities, nutritional values, and chemical composition of okara as well as its potential utilization. Owing to its peculiar rich fiber composition and low cost of production, okara might be potentially useful in the food industry as a functional ingredient or good raw material and could be used as a dietary supplement to prevent varied ailments such as prevention of diabetes, hyperlipidemia, obesity, as well as to stimulate the growth of intestinal microbes and production of microbe-derived metabolites (xenometabolites), since gut dysbiosis (imbalanced microbiota) has been implicated in the progression of several complex diseases. This review seeks to compile scientific research on the bioactive compounds in soybean residue (okara) and discuss the possible prebiotic impact of this fiber-rich residue as a functional diet on eubiosis/dysbiosis condition of the gut, as well as the consequential influence on liver and kidney functions, to facilitate a detailed knowledge base for further exploration, implementation, and development.

## 1. Introduction

Soybean is labeled as one of the essential crops across the globe and is native to Asia, and has been under cultivation for over thousands of years there. However, the present leading producers are found on the other side of the Pacific Ocean, thus north and south America [1]. America, Brazil, Argentina, and China are currently the leading world producers and consumers of soybean, respectively [2]. Epidemiological reports unveil a strong connection between regular and optimal intake of soybeans with numerous health-promoting functions, such as lowering the risk of several cancer forms (colorectal cancer, prostate cancer, breast cancer, bone health, etc.), cardiovascular diseases, cognitive function, type II diabetes, renal function, atherosclerosis, menopausal symptom, and coronary heart diseases by lowering the levels of (LDL) low-density lipoproteins [3,4,5]. Soybeans are usually processed to obtain protein isolates and other end-products like soymilk and soybean curd (tofu) [6], which are both known traditional Asian food products, however are now consumed across the globe owing to the nutritional and health-promoting claims. A large sum of fibrous residue called okara is produced after soymilk and soybean curd production processes, i.e., obtained after extraction of the aqueous fraction. Okara is a white-yellow material, which consists of the insoluble parts of the seeds remaining in the filter sack when pureed soybean seeds are filtered during soymilk production. Okara is abundant and nutritionally valuable, and has been used in the vegetarian diets of Western countries since the 20th century [7,8]. Studies on the nutritional and non-nutritional constituent of okara reveal that it is packed with a significant number of proteins, isoflavones, soluble and insoluble fibers, soyasaponins, and other mineral elements, which are all attributed with health merits [9,10,11]. Owing to its high dietary fiber content, okara supplementation yielded a decrease in body weight, beneficial properties on lipid metabolism, protected the gut environment in terms of antioxidant status, as well as prebiotic effects [8,12]. Typically, close to 1.2 kg of wet okara is obtained from 1 kg of dry soybean processed for soymilk or tofu. This makes okara a cheap source of fiber-rich food. However, it is normally used as fertilizers or is landfilled, animal feed, or discarded as waste due to its high susceptibility to spoilage, extra cost of production, undesirable flavor, and grittiness in texture attributes, which are all caused by its high moisture content. Its valorization will be essential, to aid in utilizing the untapped precious nutrients as well as serving to eliminate the socio-environmental and economic problems caused by this waste disposal [13,14,15,16]. Moreover, most of the valorization research on okara focused more on physical attributes rather than the possible health attributes [17].

In recent years, there has been a growing interest in using by-products or waste biomasses from the food industry as new sources of functional ingredients such as prebiotics. Its value addition would reduce the environmental effect from their decomposition and as well increase their benefit [18,19]. Due to the peculiar dietary fiber composition of okara, it might be potentially useful in the food industry as a functional ingredient. In this sense, it could be used to increase the dietary fiber content in varied cereal products [7,8]. Dietary fiber is evidenced to positively impact the gut health by fostering the growth of select gut microbes, as well as the production of microbe-derived metabolites (xenometabolites), and non-gastrointestinal related conditions such as non-alcoholic fatty liver disease, diabetes, and cardiovascular disease [20]. Gut dysbiosis or poor gut health has been implicated in the progression of chronic kidney disease [21]. This concept has given rise to terms such as the gut-kidney axis [22] and the gut-liver axis [23]. Nowadays, numerous studies have been undertaken aiming to uncover the functional and therapeutic effect of okara, as well as facilitate its effective utilization. Hence, this review seeks to compile scientific research on the bioactive compounds in soybean residue (okara) and discuss the possible prebiotic impact of this fiber-rich residue as a functional diet on eubiosis/dysbiosis condition of the gut, as well as the consequential influence on liver and kidney functions, to facilitate a detailed knowledge base for further exploration, implementation, and development.

### Dietary Fiber in Nutrition

The adoption of varied plant-based components including dietary fiber, polysaccharides, and prebiotics seems to be on the rise and is gaining more and more attention owing to its supposed health effects. However, there are variations in the nutritional composition of these varied fiber-containing feed/foodstuffs [24], and consumers have developed much interest in the association between the carbohydrate/fiber content of these foods and the possible glycemic effect mediated post ingestion [25]. For instance, an extruded cereal food product is regarded to reflect a high glycemic index similar to or more than that of bread, and it has been evidenced in multiple reports that an individual rate of glycemic response is dictated by the extent and rate of starch hydrolysis after ingestion and can be manipulated through daily meal dietary fiber supplementation [26,27]. Dietary fiber describes (non-digestible carbohydrates of plant origin) food-derived components that are resilient to digestion/hydrolysis by the core enzyme machinery that is present in animals/humans’ gut. Fibers are the residual parts of a plant that are safe for consumption and it includes compounds such as cellulose, lignin, cell wall polysaccharides, oligosaccharides, and other related compounds, e.g., phenolic compounds [28,29,30]. Dietary fiber is tagged as the seventh (7th) important food nutrient for organisms and is subcategorized into two types, thus, soluble dietary fiber (SDF) and insoluble dietary fiber (IDF) [13], which are all made of dense indigestible polysaccharides. The most widely acknowledged classification for dietary fiber has been to distinguish dietary components based on solubility in a buffer at a definite pH, and/or fermentability in an in vitro system with an enzyme solution serving as human alimentary enzymes. Further classification is based on fermentability such as less fermentable/water-insoluble (i.e., lignin, cellulose and hemicellulose) and well fermentable/water-soluble (i.e., gums, pectins and mucilages) [31]. Recent research proposes dietary fiber classification by size/density plots, however, the traditional and most convenient ways for dietary fiber classification still remains via solubility to water. Numerous studies have demonstrated support of dietary fiber to influence the increase in cholesterol- and sodium cholate-binding capacity as well as influence the decrease in blood pressure, protect against several cancers, including colorectal cancer, breast cancer and prostatic cancer, prevent gastrointestinal problems [32,33], ameliorate constipation (fecal softening and bulking, improve regularity and/or frequency), depicting an anti-inflammatory effect on the digestive tract as well as aid partially in lipid substitution, blood glucose regulation, and/or blood cholesterol reduction [29,34,35]. Hence, it is justifiable that dietary fiber influences the performance and functions of the gastrointestinal tract and consequently reflects on human/animals’ health [36,37]. High dietary fiber consumption is suggested to inhibit the bioavailability of some essential nutritional components including, but not limited to, vitamins and other minerals, and may impact the rate of food digestion, energy metabolism as well as gut microbial composition, which may, in turn, result in the production of short-chain fatty acid that is responsible for (10–30%) the total energy requirement of the host [37,38,39], and on the other hands, aid in the detoxification of the host digestive tract [40]. Besides, dietary fiber, which comprises polysaccharides that skips enzymatic digestion, acts essentially as a substrate to the intestinal microbiota and is suggested to impact the host-microbial community and immunity [41]. Dietary fiber deprivation in a mice experiment yielded to the alteration in mucus-eroding microbiota, disruption of the intestinal barrier, mucus layer depletion, and lethal colitis [42].

## 2. Nutritional and Anti-Nutritional Components of Soy and Soy Residue

### 2.1. Nutritional Constituents

It is well established that soybeans possess an abundant source of protein owing to their high nutritional value as well as chemical and physical properties. Besides, soybean and its by-products are evidenced in the literature as a rich source of phytochemicals/bioactive compounds, i.e., non-nutrient components of a plant with health-promoting functions and qualities. These compounds include, but are not limited to, lunasin, lectin, phytic acids, saponins, omega-3-fatty acids, phytates, trypsin inhibitors, proteins, peptides, Bowman-Birk’s protease inhibitor, phytosterols, and isoflavones, mainly daidzein, genistein, and glycitein [1,43,44]. Traditionally, all these nutritional constituents have been regarded as anti-nutrients. However, the recent advances in knowledge have yielded a better understanding of their therapeutic and beneficial health functions, from cholesterol-lowering functions to anticancer properties, diabetes-mellitus-controlling effects, and postmenopausal osteoporosis reduction [1,45].

The main components of this residue are the coating of the beans and the broken cotyledon cells [1], which are made of crude fiber, total dietary fiber, insoluble dietary fiber, and soluble dietary fiber, and are suggested in several reports to play vital roles in multiple biological processes and as well aid in the fight against syndromes of varied origins. Hence, this residue is regarded as an important source of dietary fiber owing to its major composition and low cost. However, the chemical composition is dictated by the method of soy processing or extraction, thus the quantity of water-soluble components obtained from ground soybeans and whether the residual extractable constituents have been extracted or not, and the cultivar of soybean used. Varied cultivars differ in lipid and crude protein contents, fatty acids compositions, and lipoxygenase activities [46,47]. Moreover, the variation in the nutritional profile of both wet and dry soybean residue has been attributed to cultivar differences, the incidence of sunshine, analysis methods, and the production or processing conditions used. Hence, the characteristics of water-soluble constituents may vary owing to the raw material used [48,49]. However, this lies beyond the scope of this review and, hence, interested readers can refer to these articles [10,47,50,51,52]. The sequence and procedures for processing the beans are as well very essential and it dictates the fate of all water-soluble extract in the beans. For instance, there is variation in the way the Chinese and Japanese process their soymilk and soybean curd. In the Chinese way, the soaked beans are rinsed, and the raw beans are then ground and the residue is then filtered off with water, and then heat the extract; in the Japanese system, the soaked beans are first cooked prior to grinding and filtering [7,30,46]. Figure 1 displays the schematic illustration of the steps involved in processing soy milk and the production of soybean residue/okara [46,47,52].

Although the substrates (okara) generated from soybean processing are suggested to have a high moisture content of nearly 70–80% and are mostly bound to the dietary fiber, yielding to a clumpy texture and appearance similar to wet sawdust, with mainly insoluble fiber i.e., cellulose and hemicellulose accounting for almost all it dry matter content (i.e., about 40–60%), which can be fermented by the gut microbiota in the large intestine, even though it cannot be fully digested in the small intestine. In contrast, the proportion of free carbohydrates (including, galactose, arabinose, fructose, sucrose, glucose, stachyose, and raffinose) is low (4–5%) and the lack of fermentable carbohydrate is the core factor inhibiting efficient fermentable bacterial growth in the residue. In particular, soybean residue contains 1.4% raffinose and stachyose, which may yield flatulence and bloat in some individuals. The monomers constituting the cell wall polysaccharide of the residue are mainly galacturonic acid, arabinose, glucose, galactose, fucose, xylose, and a small amount of mannose and rhamnose [53]. However, the moisture-free/dry residual content of soybean is reported to contain about 10% fats, 30% protein, and 55% total dietary fiber, thus, 5% low-soluble dietary fiber and 50% insoluble dietary fiber [48,54]. A recent study on the impact of high hydrostatic pressure (HHP) on the functionality of dietary fiber in okara was reviewed. The authors observed that subjecting HHP to soybean-derived dietary fiber increased soluble dietary fiber content (i.e., more than 8-fold), which is important in ensuring that soybean residue has anti-carcinogenic and anti-inflammatory effects on the host digestive tract [55].

Okara is suggested to be a potential source of less-priced plant protein used in human food owing to its recently confirmed high nutritive value and superior protein efficiency ratio [56]. Notably, the dry matter fraction of okara is shown to contain 15.2–33.4% protein (i.e., mainly 7S globulin and 11S globulin) [57,58]. These residual protein isolates contain all important amino acids although less soluble in water [57,59]. Again, the protein has been demonstrated to resist complete digestion by the gastrointestinal enzymes, pancreatin, and pepsin, and the latter is made of mainly steapsin, trypsin, and amylopsin. However, these low molecular weight component (less than 1 kDa) of the digestible-resistant peptides is highly potent in hindering angiotensin-converting enzyme (ACE) and hence exhibiting great antioxidant activity, possibly owing to its high fraction of hydrophobic amino acids [60]. About 5.19–14.4% of the residual protein content is made of trypsin inhibitors and can be inactivated with sufficient heat treatment [61]. Microbial bioconversion of soybean residual proteins may present few merits. Thus, its bioconversion into smaller proteins may elevate its solubility and hence generate bioactive peptides and/or amino acids. Trypsin inhibitors are suggested to be degraded by microorganisms to encourage its residual nutrient quality. However, microorganisms can catabolize residual proteins and amino acids, yielding a reduction in the number of essential amino acids present in the residual fraction. A recent study suggests that it is vital to put into account all potential effects of fermentation on the molecular weights of peptides, amino acids profile as well as the inhibitory activity of trypsin, since they play a role in impacting the overall functional characteristics including solubility and foaming properties, as well as the bioactivity of residual soy content [1,46]. Table 1 presents a summarized report on effect of heat, fungi, and bacterial treatments on soy-based products.

A study by Chan and Ma [57] reported a significant improvement in the emulsification, solubility, and foaming properties of okara protein by acid modification. The authors also discovered a distinct variation in the techno-functional properties of okara through different pre-treatments (i.e., ultrasonic, homogenization, and steam-cooking treatment), thus, the drastic improvement in the content of hydrophobic amino acids, decreasing hydrodynamic diameter, promoting surface hydrophobicity as well as enhancing solubility and oil holding capacity [73]. However, a significantly enhanced antioxidant capacity of residual protein concentrate was expressed after enzymatic hydrolysis using a combined mixture of flavourzyme and alcalase [74]. A recent study on the impact of acid precipitation (mainly HCL, malic acid, and citric acid) on the structural and functional properties of okara was thoroughly reviewed. The authors recorded a variation in the functional properties of the residual protein from soybean (mainly, 7S globulin) influenced by the acid precipitation. It was observed that citric acid yielded an increase in the size of residual protein in contrast with HCL and malic acid. HCL resulted in high solubility, foaming ability index, water holding capacity, and foaming stability index. Malic acid recorded the lowest foaming stability index, emulsifying stability index, and foaming ability index. Citric acid-induced highest emulsifying stability index and oil holding capacity. The authors concluded that acid precipitation was able to modify the functional property of the okara protein through impacting the structure, which facilitated protein extraction from the poorly soluble raw materials and hence has widened the possible application of the obtained protein in the food industry [75].

The defatted soybean residue, which is generally obtained from the production of soybean protein isolate and soy oil, is usually made of 14–25% proteins, 70–85% fibers, and less than 1% lipids [73]. The residual content of soybean is suggested to contain a considerable amount of lipids at 8.3–10.9% (dry matter). Most of the fatty acids are poly- or monosaturated and are made of linoleic acid (54.1% of total fatty acids), stearic acid (4.7%), palmitic acid (12.3%), oleic acid (20.4%), and linolenic acid (8.8%) [76]. During the grinding of soybeans, unsaturated fatty acids, mainly linoleic acid, react with soy lipoxygenase and hydroperoxide lyase yielding to the formation of aromatic compounds like hexyl and nonyl aldehydes and alcohols. These formed odors with low detection thresholds signify the aromas/off-flavors in raw soymilk. Since these enzymes are generally denatured at a temperature above 80 °C, the Chinese way of processing soymilk (i.e., the raw soybeans are ground before the filtrate is heated) is likely to generate residue with greener character and beany flavor [77]. Hence, the variant obtained using the Japanese way of soymilk processing is relatively more palatable and is likely to have a lower content of trypsin inhibitor, thus it can easily be reused during cooking and processing [61]. However, this may define the reason why soybean okara is common in the Japanese market but rarely found in the Chinese markets. Fermentative microorganisms could metabolize fatty acids and their respective derivatives to produce much desirable aroma compounds. A recent study on the recovery of okara oil components through supercritical carbon dioxide extraction modified with ethanol indicated that at a pressure of 20 MPa and a relatively low temperature of 40 °C in the presence of 10% mol EtOH yielded to the recovery of about 63.5% oil-component. The obtained oil component was made of phytosterols, fatty acids, and traces of decadienals. EtOH retained its dignity by elevating the yield and composition of phenolic compounds in the extracts, mainly soy isoflavones (i.e., genistein and daidzein). Soy isoflavones are well-known antioxidants that can increase both the value and stability of the oil, making the process attractive for food, cosmetics, and even in the pharmaceutical industries [78]. On the other hand, soybean residue is evidenced to contain a variety of minerals, a fair amount of iron, calcium, and potassium [53,79].

### 2.2. Anti-Nutritional/Bioactive Constituents of Soybean Co-Products; with Emphasis on Polyphenols (Soy Isoflavones)

Soy food such as soymilk contains a combination of balanced nutrients that is comparable to that of cow milk, but with gluten and lactose, and is embedded with promising phytochemical compounds that are linked with health-promoting functions. Soy foods and products are evidenced in several reports to possess a relatively high and diverse group of phenolic compounds, including phenolic acids, flavonoids, and non-flavonoids. Their vital role in our daily diet as bioactive compounds have been widely explored, with growing evidence revealing their role in decreasing chronic disease risks, such as cardiovascular diseases, diabetes, immune dysfunction, age-related eye problems, and cancers, which are all associated with the antioxidative effects of these phenolic compounds [80]. Bioactive compounds are molecules evidenced to portray therapeutic potentials with an impact on the metabolic disorder, energy intake, oxidative stress, as well as reducing proinflammatory state [81]. The main bioactive components of soybean are proteins or peptides, saponins, phytosterols, isoflavones, protease inhibitors [82,83], tocopherols, and carotenoids [84]. Vong and Liu [46] reported on biologically active constituents of okara and they include acetyl glucosides (0.32%), saponins (0.10%), phytic acid (0.5–1.2%), malonyl glucosides (19.7%), isoflavone aglycones (5.41%), and isoflavone glucosides (10.3%). Past research has revealed that soybean is rich in polyphenol mainly isoflavones. Soy isoflavones are regarded to depict essential biochemical properties as part of the flavone compounds. Its role as an estrogen-like plant chemical (phytoestrogens) [85], has made them a topic of much interest and have been under surveillance by researchers, as they are accredited with important activities against hormone-derived cancers, menopausal syndrome disorders, osteoporosis [86,87,88], blood cholesterol, cardiovascular syndrome, and cognitive function [89]. Isoflavones are well-known polyphenols with a chemical structure similar to that of flavones. Both isoflavones and flavones are subclasses of flavonoids, which are in the largest polyphenolic groups [81,90,91,92]. Soybean contains up to 12 varied categories of isoflavones, which can be segregated into three (3) main groups (i.e., glycitein, genistein, and daidzein), all of which can take four varied forms: β-glucosidase, aglycones, malonyl-glucosides, and acetyl-glucosides, which constitute the main phenolic components and have been attributed to performing many health-promoting functions [86,89]. A recent study on the composition of okara was reviewed, and the authors reported that the total isoflavones content of okara is 355 mg/g (dry weight basis). The concentration of aglycones, malonyl glucosides, isoflavone glucosides, and acetyl glucosides in the residue were found to be 54.1, 196.8, 103.2, and 3.2 mg/g, respectively [89]. As mentioned above, okara may contain the same 12 isoflavones, although the processing conditions during soymilk production may impact the original isoflavones profile [86]. Another factor that can impact the profile of isoflavone in okara is their associations with other food matrix components, including non-covalent interactions between macronutrients and polyphenols, mainly proteins [81,93,94]. However, approximately 12–30% of the isoflavones contained in soybeans are suggested to be retained in the residue during soymilk processing. The main residual constituent of soy isoflavones is aglycones (15.4%), glucosides (28.9%), and a small fraction of acetyl genistin (0.89%) [95]. β-glucosides and malonyl-glucosides are the basic forms in soybeans, which can be transformed into acetyl glucosides and aglycones during processing due to thermal stress or enzymatic conversion [96]. The study by Izumi et al. [97], on the absorption rate of soy isoflavone aglycones in a human, was reviewed. The authors reported that β-glucosidase may enzymatically hydrolyze isoflavone glucosides into their aglycone forms, depicting a greater bioavailability in humans. Additionally, selected fermentative microorganisms are evidenced by experts to secrete β-glucosidase, hence the bioconversion of isoflavone glucosides in soy residue into aglycones via fermentation provides opportunity for further value-addition [98].

The health effect of isoflavones, including anti-inflammatory and anticancer properties, cardiovascular defenses, as well as enzyme inhibitory roles of isoflavone are mainly linked with their antioxidant capacity, which is comparable to or better than that of other polyphenols [85,92]. These health effects have as well depicted to be useful against type 1 and type 2 diabetes mellitus, and have been widely proven in numerous reports [56]. An antioxidant is classified as an organic compound, which when available in small concentration/quantity in contrast with an oxidizing substrate, can significantly combat that substrate’s oxidation [92]. Although the term officially defines compounds that react with oxygen, it can also adhere to compounds that guard and/or protect against free radicals (i.e., molecules with unpaired electrons) that deny these radicals from harming healthy cells [47]. Daidzein and genistein are the strongest soy isoflavones for antioxidant activity. Genistin is evidenced in numerous reports to protects against oxidative DNA damage yielding from hydroxyl radicals, such as superoxide anion scavenging ability [56] as well as prevention of low-density lipoprotein oxidation [99]. The research on the impact of different storage conditions and thermal treatments on the stability of okara revealed genistin as the most dominant residual glucoside (0.33 mg/g) along with daidzin (0.25 mg/g), genistin (0.32 mg/g), genistein (0.02 mg/g), and daidzein (0.02 mg/g) of dried okara in a high-performance liquid chromatography study [100]. However, the exploitation of soy residue/by-products for the recovery of bioactive compounds has aroused much interest focusing on the contribution to food production and sustainable agriculture [101]. In fact, these by-products from soybean are often very rich in phenolic compounds, owing to their presence in the seeds and peels, which are often retained in the residues. Their tendency and relatively low water solubility to associate with other components may impact these by-products with rich polyphenolic content. Numerous potential applications of phenolic compounds have been reported, such as antioxidant stabilizers, food flavorings and colors as well as bioactive ingredients for health. Several unconventional and conventional techniques have been suggested for the separation of these high-value components. The extraction for conventional solid-liquid commonly uses hydroalcoholic mixtures [102]. Moreover, many other solvents, including acetonitrile, acetone, ethyl acetate, and methanol, are still under intensive study in the extraction of polyphenols, owing to the relatively easy solubilization portrayed by these solvents and mixtures [103]. Alkaline, acid, and sub- or supercritical fluid extraction are known alternatives. Modern technologies, including pulsed electric fields, microwave-assisted extraction, and ultrasound-assisted extraction have been proposed as means to encourage yields as well as overcome some challenges in polyphenol extraction. Examples of possible difficulties include component instability and solvent residues in the final product, as well as the kinetic limitations in cell-matrix extractions [1].

The preparation of soy molasses, i.e., a by-product of soy-protein concentrate, is a common starting material for the production of isoflavone. Being a known alcoholic extract of soy flakes, it is embedded with isoflavones in a slightly more concentrated form. Nevertheless, many patented processes use soybean and soybean meal as a commencing material during the recovery of isoflavones from side products, such as okara, which would require fewer valuable resources. Table 2 presents the general nutritional constituents of okara [46].

## 3. Valorization of Soybean Residue (Okara) as a Functional Food

As discussed earlier, okara contains high levels of dietary fiber and proteins, and significant amounts of isoflavones as well as mineral elements, which merits it a high nutritional value and a potential prebiotic function. Hence, it is potentially useful as a functional ingredient with health-promoting effects [8]. Specifically, the incorporation of soybean-derived ingredients into a variety of products with the motive of providing beneficial properties and functions to the body has been the focus over the years, and have aroused much interest from the food industry. These are termed ‘functional foods’ [104]. Okara has been used in food production for human consumption as well as application in animal nutrition for several years mainly in Japan and China, both in its processed and raw forms, to more easily provide a fair intake of the nutritional claim for the fiber and protein. Okara can be a partial substitute for soy flour, wheat flour, and other food-producing components so as to boost the protein and fiber content [49]. The rich amount of proteins, carbohydrates, and other forms of nutrients embedded in okara make it a potential substrate for microbial fermentation. Bacteria, yeasts, and fungal fermentation of soybean residue are suggested to decrease the content of raw fiber, increase the content of proteins, soluble fiber, isoflavones, and amino acids, and decomposing phytic acid, resulting to an upgrade in the processing properties as well as nutritional value [105]. The use of soybean residue in different food formulation such as drinks, bread, sausages, pancakes, candies, biscuits, cake, and nutritional flour, has been studied and evidenced earlier in numerous reports [15,106,107,108,109].

### 3.1. Application of Soybean Residue in Human Nutrition

The rich solvent-binding properties of okara make it an ideal low-cost ingredient with which to encourage yields in bakery and meat products [48]. It has been evidenced to have a positive influence on the shelf life of chocolate cookies mainly at an optimal concentration of 5%, and as well prevent syneresis in cheese ravioli filling during defrosting and freezing. Besides, its insipid taste enables it to be used at relatively high levels without adversely influencing the texture or taste profiles of the formed products [1].

The study by Park et al. [110], on the influence of okara and additives including, soy flour, starch, and hydroxypropyl methylcellulose on the quality and nutritional value of cookies were reviewed. The results displayed that okara-enriched cookies had higher levels of carbohydrate (35.3%), fat (25.7%), protein (11.6%), and ash (6.3%) in contrast with wheat cookies containing carbohydrate (59.6%), fat (20.2%), protein (15.2%), and ash (2.2%). A cookie containing okara depicted lower carbohydrate and higher ash content. Hydroxypropyl methylcellulose supplemented cookies portrayed a higher water holding capacity, i.e., three times higher than that of control, which enhanced the performance of the dough and the quality of enriched cookies. Cookies supplemented with additives, mainly, soy flour and hydroxypropyl methylcellulose, portrayed reduced water activities, which enhanced the storage life and hardness in consort with significant improvement in the crispness of okara-enriched cookies. Suda et al. [111] incorporated okara powder (containing 50% dietary fiber, 0.45% calcium, and 21.3% vegetable protein) into bread and pancake ingredients, intending to develop fortified foods for medical use. Three dissimilar types of bread were processed, which include the 10% okara bread, and other additives, as well as preservatives, were needed to encourage yeast fermentation and enable storage at room temperature. After effective freezing without preservatives, the flavors of all three bread were altered. A soft pancake was prepared using a mixture of pancake powders with 20% okara. The pancake free from preservatives was suitable for refrigerator storage before being eaten, while both the fresh pancakes and those comprising the preservatives made with 20% okara were accepted as supplements for the elderly hospitalized patients. The study showed that the soft pancake supplemented with okara and 40% water was much use as a fitting supplement for dietary fiber, calcium, and vegetable protein than the bread supplemented with okara. The quality of okara supplemented bread is suggested to significantly increased with the addition of enzymes (lipase, glucose oxidase, and pentosanase). The incorporation of 4–8% soybean powder has been evidenced to yield a quality increase [112]. The application of 5% soybean dietary fiber on bread, treated with 1% NaOH for an hour and 1% HCl at 60 °C for 2 h, depicted an appearance and quality similar to that of a normal bread [1,113].

A recent study by Kang et al. [15] on the quality attributes of rice noodles enriched with different levels of okara flour (0–20%) was reviewed. The authors recorded that the adhesiveness, hardness, and cooking loss of the noodles increase with an increasing amount of okara, while swelling index, cohesiveness score, and water absorption significantly decreased. Among all samples, noodles fortified with 10% okara flour demonstrated the lowest score for the predicted glycemic index. The incorporation of alginate with a CaCl_2_ coating upgraded the cooking properties without affecting the in vitro starch digestibility of soybean residue-enriched rice noodles. The findings recommended that noodles with good quality attributes and reduced in vitro starch digestibility scores could be developed through incorporating okara flour of up to 10% level. Again, a study on the cooking quality of soybean by-product okara fiber fortified noodle reported that the noodles were of good cooking quality when okara fiber was supplemented at 9% (particle size of 100 mesh), 4% salt, and 0.25% sodium alginate [114]. Pan et al. [106] studied the impact of okara and vital gluten on the physical and chemical properties of noodles. The results portrayed that a higher level of okara (10–15%) significantly decreased the tensile strength, extensibility, elasticity, and optimum cooking time of the noodles. Noodles enriched with okara displayed an improved flavonoids content, total phenolic, and antioxidant and/or free radical-scavenging activity. The results showed that 6% wheat gluten and 5–10% okara powder incorporation produces noodles with good texture, cooking, and sensory properties. Hence, soybean dietary fiber supplementation in food formulation can be the best alternative in achieving food products with a low glycemic index.

In the study by Katayama and Wilson [115], a soybean-based snack was formulated using partially dried okara (44.3% moisture) prepared from lipoxygenase-free soybeans and a commercially dried okara powder (7.7% moisture) made from regular soybeans (lipoxygenase-present). The authors also utilized commercially low linolenic acid soybean oil and commercially low saturated soybean oil in the same recipe and determine the best recipe for baked or a deep-fried soy-based food product. At the end of the study, the authors observed that the baked foods made from the partially dried okara-free lipoxygenase powder and the commercially low-saturated soybean oil, had a taste, appearance, and texture parallel to the reference product (i.e., commercial Japanese okara-based snack). The final product had 11.4% protein and 7.4% dietary fiber, which was 2.0 and 1.5 times higher in contrast with the standard. The amount of calcium was as well higher (4.3 times) than the reference score. Bedani et al. [116] also recorded an increase in the functional and nutritional properties of soy yogurt incorporated with soybean by-product okara. All these findings suggest the commercialization of okara at the industrial level as a potential nutritionally enhanced value-added food product.

### 3.2. Application of Soybean Residue in Animal Nutrition

Owing to the higher non-fibrous carbohydrates and proteins content of okara, as well as being cheaper than soybean meal, makes it attractive to be used as feed for dairy cattle, goats, sheep, pigs, poultry, and fish, with the aim of substituting part of their normal feed [117,118,119,120].

The by-products of food crops are generally known to possess high moisture content and, the avoidance of the high energy costs of drying, are hence usually stored by silage, a forage storage technique that involves acidifying the plant mass via anaerobic microorganisms, to aid minimize and deny it from being colonized by other microorganisms that will result to the loss of its nutritional value. A famous practice in Japan, with the aim of preventing nutritional loss, curbs this by mixing dry feed with wet by-products to achieve low moisture total mixed ration silage. The mixture of peanut hulls and okara can therefore have a synergistic influence on the silage mix to form the right dry matter fermentable carbohydrates for ideal silage fermentation. Silage with peanut hulls/okara ratio 22:78 has been evidenced to decrease fiber content and lignification, as well as improve the efficiency of both in-vitro ruminal fermentation and silage fermentation models after 8 weeks [117].

Meeting the required protein profile in young pigs especially for organic pig producers is a challenge. Thus, the price of organic feed is significantly higher (i.e., 4 times higher), and has limited availability. Okara is suggested as an alternative and a potential source of organic protein, and its intake in up to 25% of young pigs’ diets depicted no effect on average daily food intake, average daily profit, and gain/feed ratio in contrast with the control [121]. Wang et al. [120,122] also substituted dried okara for soybean meal in dairy cattle and yellow cattle for a duration of 30 days. The authors recorded no significant differences in the production of milk, feed consumption, milk-fat content, and daily profit between both groups. However, the feeding cost of the group substituted with dried okara was significantly reduced.

In addition, okara can also be used in the production of microbial proteins, which are synthesized via solid-state fermentation. During the process of fermentation, the mold degrades the residual fiber into low molecular weight carbohydrates, which are well along used by yeasts to synthesize proteins. Moreover, some anti-nutritional factors (including, but not limited to, trypsin inhibitors, lectin, and saponin) can be further decomposed or reduced through fermentation [123]. Pan et al. [124] studied the changes in microbial production via mixed culture solid-state fermentation on okara. The authors observed a doubled crude protein content in contrast with the original material when using wheat bran and okara (ratio 2:8) as substrate, and *Trichoderma viride*, *Aspergillus niger*, *Saccharomyces cerevisiae*, and *Candida utilis* (1:1:1:3) as a mixed crop, after fermentation for three days at 32 °C. Table 3 and Table 4 presents summarized studies on the effect of okara supplementation on nutrition, as well as the impact of its fortification on food properties and function respectively (in vivo/in vitro).

## 4. Physicochemical and Prebiotic Influence of Dietary Fiber (Emphasis on Okara-Derived Fiber) on Gut and Associated Tissues: Gut, Liver, and Kidneys

Numerous shreds of evidence are indicating that the gut microorganisms can directly influence the physiological conditions in human/animals such as improving the host immune system, improving the roles of the intestinal barrier, stimulating the defensive mechanisms against pathogens as well as increasing the defensive mechanisms against inflammatory bowel diseases, producing biological metabolites, regulating autoimmunity, regulating diabetes and preventing obesity, and destroying cancer cells. The host gut–microbiota interactions are dynamic and highly dictated by several environmental conditions particularly diet [135]. In addition, the gut is reported to have dual and opposing roles of enabling nutrients to enter the body while denying the entry of harmful substances. Both gastrointestinal barrier function and nutrient absorption have been demonstrated to be altered by dietary fiber. For instance, dietary fiber-induced changes to the gastrointestinal barrier by increasing mucins and the cell that produce them “goblet cells” [136,137]. Mucins are large glycoproteins that, along with lipids, antibodies, bacteria, ions, proteins, antimicrobial peptides, and water, form what is termed as mucus [138]. Mucus protects the gut epithelium from mechanical stress, to prevent the translocation of harmful substances as well as to lubricate the intestine and to facilitate easy transportation of digested material. A study comparing a standard rodent chow/diet (i.e., fiber from corn, oats, and wheat comprising 4.3% of the diet by weight) with a chow devoid of any fiber displayed that mice fed with a fiber-deficient diet had a thinner mucus layer, hence enabling microbes to get in closer proximity to the gastrointestinal epithelium [139]. Therefore, insufficient amounts of dietary fiber in the gut may encourage bacteria to degrade the host mucus layer (i.e., breaking down one of the host’s physical barriers) in an attempt to provide themselves with substrates needed to survive.

### 4.1. Physicochemical Role of Dietary Fiber in The Gastrointestinal Tract

The main role of the gastrointestinal tract is the absorption of nutrients from ingested foods. This absorption is preceded by a series of digestive processes within the different gut compartments. These processes are governed by the secretion of enzymes and associated co-factors, as well as through maintenance of gut lumen at optimal pH conditions for digestion. [140]. Classically, the consumption of dietary fiber has been postulated to affect the uptake of nutrients in diverse methods. The physicochemical factors of dietary fiber, including fermentation, bulking ability, viscosity and gel formation, binding ability, solubility, and water-holding capacity, have been proved to influence nutrient absorption. A plethora of in vivo and in vitro studies have been carried out in the past decades to aid illuminate the physicochemical interactions between dietary fiber and these nutrients [39]. However, it appears that water-soluble dietary fiber, which is either viscous or gel-forming under intestinal/gastric conditions reduces the rate of absorption than low molecular weight and low viscous fibers. The small intestine is the main absorptive area in the gut, which, per the dietary stand-point, involves in the absorption of the subunits of digestible macronutrients (i.e., monosaccharides from carbohydrates, amino acids and some di/tripeptides from proteins, and fatty acids/glycerol from di/triglycerides), as well as minerals and vitamins, and other micronutrients [140].

Dietary fiber differs in form depending on their biochemical and physiological properties, and hence affect the bioavailability of nutrients, composition of microbiota and gastrointestinal functions. Soluble dietary fiber portrayed an increased viscosity and reduced starch digestibility [141], as well as suppressed the activity of α-amylase in the gastrointestinal tract, which consequently reduced the rapid rise in postprandial blood glucose levels [142,143]. The consumption of soluble viscous fibers, together with other dietary changes like reduced fat intake, is effective in reducing cholesterol levels [144]. Sasaki and Kohyama [145] studied the influence of various soluble dietary fiber on starch digestibility. They recorded a strong connection between apparent viscosity at low shear rates and inhibition of starch digestibility. However, insoluble dietary fiber decreased the digestibility of starch by non-specific enzyme adsorption. Nagano et al. [146] systematically reviewed the improved functional and physicochemical properties of okara, by nanocellulose technologies used in food development. The authors hypothesized that nanocellulose technologies may improve physicochemical roles, hence influencing the gut microbiota community. Their study demonstrated an increased viscosity, dispersion ability, as well as the specific surface area of cellulose and okara. The increased viscosity triggered the suppression of α-amylase activity whereas increased dispersion ability and specific surface area of okara resulted in improved SCFA production of human dominant gut bacteria [146]. *Bifidobacteria* and *lactobacilli* are the most important health-promoting bacteria of the gut microbiota. Hence, both are common targets for dietary interventions to promote health. Other bacteria including *enterococci*, *eubacteria streptococci,* and *Bacteroides* can be classified as potentially harmful or beneficial to health depending on the species [126,147]. The healthy bacteria exert beneficial functions to the host through their metabolisms such as SCFA formation (mainly propionate, acetate, and butyrate), absence of toxin production as well as the synthesis of defensins or vitamins [126]. Desai et al. [42] researched the consequential impact of dietary fiber deprivation on the gut microbiota using a gnotobiotic mouse model. The authors observed that during intermittent or chronic dietary fiber deficiency, the gut microbiota turns to the host’s secreted mucus glycoproteins as a nutrient source, consequently yielding to an erosion of the colonic mucus barrier. Thereby, promoting greater epithelial access and lethal colitis by the mucosal pathogen, *Citrobacter rodentium*. The study disclosed intricate pathways linking diet, the gut microbiota, and intestinal barrier dysfunction, which could be ameliorated via dietary therapeutics.

### 4.2. The Use of Soybean Residue as a Prebiotic

The consumption of okara as prebiotics/non-digestible food components can selectively stimulate the composition and/or activity of one or a limited number of gastrointestinal microbes, that can confer health merits to the host [148], as well as non-gastrointestinal related conditions such as cardiovascular disease [149], chronic kidney disease (CDK) [150], non-alcoholic fatty liver disease (NAFLD) [21], and diabetes [151]. The prebiotic influence of okara has been researched in in-vitro studies using *Lactobacillus acidophilus* and *Bifidobacterium bifidum* [152,153]. Okara provided a surface for bacteria cell adhesion, thereby enabling substrate uptake as well as cell growth. The degree of fermentation of the residue was about 3.6 times more in *B. bifidum* compared with *L. acidophilus* [152]. Treatment with β-glucanase (Ultraflo L^®^) improved the soluble dietary fiber content in okara, subsequently enhancing its fermentation by *B. bifidum* [154]. The conversion of okara insoluble dietary fibers to soluble fibers was again observed when *Streptococcus thermophilus* and *Lactobacillus delbrueckii* subspecies *bulgaricus* were both used [155].

There are numerous studies on the dietary effects of yogurt fortified with soybean residue/okara, produced by lactic fermentation of soymilk and soy milk residue, on lipid and cholesterol levels in rats [156]. Okara fortified yogurt was prepared by mixing dried okara with soymilk, in a ratio of 1:2, and the mixture was then fermented with *L. delbrueckii* subspecies *delbrueckii.* The formed product was freeze-dried and then incorporated into the rat’s diets. Regardless of their diet, rats fed with dried okara and soymilk yogurt portrayed a significant and consistent lower level of total plasmatic cholesterol in contrast with the control group and another group fed with only soymilk yogurt. The authors concluded that the use of okara provided additional merits to soymilk, thus the fiber-rich residue facilitated the excretion of bile acids via their absorption to fecal matter, therefore encouraging the hypocholesterolemic effect [156,157].

Furthermore, fermentation is suggested to play a crucial role in the cholesterol-lowering effect as it triggered the production of bioactive peptides through the enzymatic hydrolysis of soy proteins [156]. The DNA microarray analyses result also confirmed that the consumption of soybean residue and soymilk yogurt reduced the synthesis of lipids and cholesterol, and up-regulated the β-oxidation of fatty acids and cholesterol disintegration [157]. In summary, these studies demonstrated that both the supplementation of okara and its subsequent fermentation in a soymilk yogurt matrix depicted hypocholesterolemic effects.

The organoleptic qualities and textural profiles of okara fortified yogurt were then assessed. Soymilk with only dried soybean residue or with insulin and dried okara was fermented with a commercial yogurt starter culture that contained *L. acidophilus*, *Bifidobacterium animalis* subspecies *lactis,* and *Streptococcus thermophilus* [116]. These yogurts exhibited significantly greater physical stability and were ranked lower in hedonic tests, probably owing to the relatively large size of the dried okara. However, the addition of insulin seems to increase palatability, thus the acceptance score was the greatest for the yogurt with both insulin and okara added.

### 4.3. Prebiotic Status of Soybean Residue on Gut Microbiome

Research on human gastrointestinal microbiota labeled as the “second brain”, or “forgotten organ” has increased exponentially in recent years following the latest advances in technology [158]. The gut is the core immune organ in the body, which harbors approximately 70–80% of the body’s immune cells and has been tagged as the largest source of inflammation evidenced to contribute to diseases like NAFLD and CDK [20,159,160]. It has been proved that the gut microbiota produces not only metabolites that can impact host physiology, but rather with metabolites that can also contribute an essential role in the host’s immune system and metabolism via a complex set of chemical interactions as well as signaling pathways [161,162,163].

Prebiotics are non-digestible parts of food termed carbohydrates that act as fibers. Unaltered, they reach the colon where they are used by the gastrointestinal microorganisms, they serve as food for “good” intestinal bacteria, and encourage their growth, colonization as well as sustainability in the digestive tract. Among prebiotics or fiber types, are the most important galactooligosaccharides, and oligosaccharides, which are referred to as bifidogenic substances; relating to their ability to selectively enhance the growth of *Bifidobacterium* spp. *(B. breve, B. longum, B. infantis, B. pseudolongum, B. lactis*) and *Lactobacillus* spp. (*L. plantarum, L. casei, L. reuteri, L. acidophilus*) [164]. The most commonly used prebiotics in human studies include, but are not limited to, Fructooligosaccharide, Galactooligosaccharide, Xylo-oligosaccharide, Arabinoxylan-oligosaccharides, and Soybean oligosaccharides [164,165].

Pérez-López et al. [51] studied the prebiotic effect of okara on gastrointestinal microbiota in high-fat-fed rats. To improve the solubility of the okara and achieving a soluble dietary fiber-enriched residue, the authors applied high hydrostatic pressure treatment together with the Ultraflo^®^ L enzyme in the preparation. Wistar rats were fed with a high-fat diet supplemented with 20% treated okara for four weeks. The authors observed a reduction in plasma triglycerides (1.4-times), weight loss, and an enhancement in the metabolism of amino-acid (1.2-times lower urea) in contrast with the control group. Regarding its prebiotic status, okara enhanced the release of short-chain fatty acids (SCFAs) and improved magnesium and calcium absorption. Furthermore, quantitative PCR analysis of selected bacterial groups affirmed that okara protected the values of several beneficial groups in gastrointestinal microbiota and restored dysbiosis caused by a high-fat diet. Thus, preventing bacterial drop triggered by a high-fat diet. The authors concluded that the supplementation of okara exerted health-promoting functions in vivo and hence could be used as a prebiotic as well as a functional ingredient in foods.

Villanueva et al. [12] as well studied the effect of high-fat diets supplemented with 13% or 20% okara on lipid profiles of the liver, plasma, and feces in male golden Syrian hamsters after three (3) weeks of feeding. The diet did not yield any significant differences in body weight gain or feed intake (*p* > 0.05). However, plasma levels of triglycerides, total cholesterol, and VLDL- plus LDL cholesterol in hamsters fed with 20% okara decreased significantly (*p* < 0.05) compared with the control group. Though, no significant differences (*p* > 0.05) were observed in HDL- and LDL-cholesterol plasma levels in all the experimental groups. Triglycerides, total lipids, total and esterified cholesterol concentrations in the liver were decreased by 20% okara supplemented diet. Regarding hamsters fed with 13% okara fortified diets, the mean values of the triglyceride, total lipid, and cholesterol in the liver and plasma reduced as compared to the control group, but the differences were not statistically significant. Besides, all assayed okara supplemented diets depicted increased fecal excretion of total lipids, free cholesterol, triglycerides, and total nitrogen (*p* < 0.05) in contrast with their respective controls. The authors suggested that the main components of okara, i.e., dietary fiber and protein, could be attributed with the total lipids and cholesterol decreased in the liver and plasma, and the fecal output increase in high-fat-fed hamsters, and hence might play an essential role in the prevention of hyperlipidemia and could as well be used as value-added ingredient for functional food preparation.

In 2016, Villanueva et al. studied the potential prebiotic and fat-lowering effects of enzymatically treated okara in high-cholesterol-fed Wistar rats [6]. The authors observed a significant reduction in the serum and liver triglyceride levels (*p* < 0.01) of rats fed with enzymatically treated okara. Total lipids, bile acids, and triglycerides were significantly (*p* < 0.001) higher in the feces of rats fed with enzymatically treated okara diet. However, the pH of fecal contents was reduced (*p* < 0.001), probably owing to the significant increase in the production of short-chain fatty acids in the treated group compared to the control group (i.e., Total SCFA (mmol/g) of 229.2 ± 37.0 for the control group and 568.2 ± 56.5 for the treated group). In addition, the enzymatically treated okara yielded reduced triglycerides in liver and serum, and improved the excretion of total lipids, bile acids, and triglycerides, increasing the lipid profile in high cholesterol-fed rats. The fiber content is suggested to aid improve intestinal transit via increasing fecal bulk. The authors concluded that the reduced pH and improved SCFA production affirmed the occurrence of fiber fermentation, hence signifying a potential prebiotic effect.

### 4.4. Gut, Liver, and Kidney Responses to the Prebiotic Effects of Dietary Fiber in General

In the microbial process of prebiotic fermentation in the large intestine, SCFAs such as butyrate, propionic acid, acetic acid, vitamin K, and vitamin B12 are formed, which are then absorbed by the gastrointestinal mucosa and distributed via the lymphatic and vascular system to cells of an organism [166]. The resulting butyrate is metabolized directly in the gastrointestinal epithelium, where it acts as a regulator of cell division and growth. Propionate is used in the liver and as well serves as a precursor used for suppressing cholesterol synthesis. Acetate is predominantly metabolized in the muscle cells, heart, kidneys, and brain. The production of SCFA decreases the pH of the environment, hence encouraging cell differentiation and growth of intestinal epithelial cells as well as re-supporting the microflora. In addition, fermentation in the intestine is suggested to produce the final form of the degradation of substances like simple gases: Carbon dioxide, hydrogen, methane, and hydrogen sulfide [164]. Figure 2 presents the schematic overview of the major prebiotic effects of dietary fiber on the gut, liver, and kidney [20].

#### 4.4.1. Gut Responses

The increase in the production of short-chain fatty acids (SCFAs) resulting from the fermentation of dietary fiber has been shown to boost gut barrier function via increasing gut cell proliferation and diversity [20,167]. SCFAs decrease gastrointestinal pH, which may alter the gut microbiota by obstructing the growth of pathogens and decrease the expression of microbial virulence genes [168]. Again, it has been evidenced that epithelial cell lines metabolize the SCFA butyrate, yielding oxygen reduction that results in the stabilization of transcription factor, hypoxia-inducible factor-1 alpha (Hif-1α) [169]. This transcription factor has been implicated in gastrointestinal barrier function via regulating apoptosis [170] and inflammation [171]. A microarray study confirms that high-amylose resistant starch could improve hormones as well as structure and function of the gut via increasing the levels of Hif-1α expression together with genes related to cell growth, differentiation, apoptosis, and proliferation in the cecal tissue of rats supplemented with 30% resistant starch in contrast with rats fed an equal amount of energy from a low-fiber diet [172]. A recent study by Zhang et al. [173] evaluated the effects of prebiotics on bioavailability and biotransformation of ginsenosides in rats after a two-week prebiotic (i.e., galactooligosaccharide, fructooligosaccharide, and fibersol-2) intervention. The results displayed that area under concentration-time curve and peak plasma concentration of ginsenoside and its intermediate metabolites in the prebiotic intervention groups were improved at various degrees than those in the control group. In addition, the gut microbiota significantly responded to the prebiotic treatment at both functional and taxonomical levels. The gut metagenomic analysis as well unveiled the functional gene amelioration for polyketide/terpenoid metabolism, gluconeogenesis, glycolysis, and propanoate metabolism, etc. The authors concluded that prebiotics may selectively endorse the proliferation of certain bacterial stains having glycoside hydrolysis capacity, hence, subsequently encouraging bioavailability and biotransformation of primary ginsenosides in vivo.

The tight junction proteins are another key component of the gut barrier affected by dietary fiber. A study by Cani et al. [174] observed that feeding a standard rodent chow supplemented with (10%) fructooligosaccharides improved the expression of genes in the jejunal tight junction proteins and Zonula occludens-1 (ZO-1), decreased intestinal permeability, and reduced plasma lipopolysaccharides (LPS) concentrations. Again, Fukunaga et al. [175] studied the effect of soluble fiber pectin on the production of fecal short-chain fatty acid, intestinal cell proliferation, and microbial population. The results showed that rats fed 2.5% pectin for two weeks displayed increased plasma glucagon-like peptide-2 (GLP-2) and increased cecal SCFAs. Butyrate, in another Caco-2 cell culture model, was again demonstrated to activate AMP-activated protein kinase, enabling tight junction protein assembly as well as improved barrier function shown via improved transepithelial electrical resistance [176]. Ohata et al. [177], also used the Caco-2 cells culture model and observed that butyrate improved lipoxygenase activity by hindering histone deacetylation, which caused transepithelial electrical resistance to increase. Apart from the alteration of intestinal immune function and physical barriers to minimize injury from microbe-derived proinflammatory factors, prebiotic can as well protect organs like the kidney and liver from metabolic abuses. It has long been evidenced that constant consumption of nondigestible carbohydrates instead of digestible carbohydrates, minimizes the increase in blood glucose and insulin [20]. Mithieux et al. [178] researched intestinal gluconeogenesis, which is another carbohydrate regulatory pathway influenced by dietary fiber consumption. The gastrointestinal production of glucose is understood to improve glucose sensing in the portal vein, resulting in reductions of hepatic glucose production as well as influenced signaling to the brain, yielding an increase in satiety. However, microbiota-generated metabolites of prebiotic (e.g., propionate), are suggested to act as a gluconeogenic precursor [179]. In summary, this highlights the merits of microbe-derived SCFAs in supporting the physical components of the gut barrier (mucosal layer, tight junctions, and cellularity) and influencing host immune factors, which will consequentially impact functions of tissues such as the liver and kidneys.

#### 4.4.2. Liver Responses

Indeed, it is known that blood from the gut reaches the liver via the portal vein, and hence is justifiable that this organ is a target of gut-derived factors altered by diet and microbiome changes. However, prebiotic fiber is being considered an ideal treatment and management option for NAFLD and related obesity and insulin resistance [21]. The hepatic effects of prebiotic fiber as discussed earlier include influencing gut ecology, thus gut permeability, systemic inflammation, as well as circulating gastrointestinal-derived hormone and metabolite signals. Endorsing the association between gut health and liver, patients with NAFLD were seen to portray an altered gut microbiota [159] and improved gut permeability [180]. Dietary fibers have been demonstrated to hinder the translocation of bacterial products such as LPS [181]. Thus, this will aid to decrease hepatic exposure to LPS and other associated microbe-derived proinflammatory products. This may hinder the tendency of fatty liver progressing to the inflammatory form termed as non-alcoholic steatohepatitis (NASH) [20]. This transition is thought to occur in two stages known as the “two-hit” hypothesis. The “first hit” is the state where the liver is susceptible to metabolic insults or the “second hit”, owing to liver fat accumulation. The “second hit” is assumed to come from a variety of sources, such as bacterial overgrowth, and is suggested to induces hepatic inflammation [182]. Cortez-Pinto et al. [183] evaluated the effect of different dietary patterns in NASH patients. The study revealed that these patients consumed low fiber, more fat, and less carbohydrate than the healthy controls. In addition, another study discussed the therapeutic effects of whole grains derived dietary fiber and demonstrated its merits associated with metabolic syndrome (MtS) and NAFLD after consumption. Besides the reduction of liver fat, whole grains derived dietary fiber could as well reduce inflammation [184,185,186].

The proposed effects of prebiotic fibers on lipid metabolism are attributed to inhibition of de novo fatty acid synthesis (FAS) [187,188,189]. Prebiotic fibers are thought to downregulate hepatic lipogenic enzymes, mainly FAS, by increasing the production of SCFA propionate [190]. Again, due to their unique modes of action regarding cholesterol metabolism specifically linked with their fermentability and modulation of the microflora. As reported earlier, SCFA has been implicated in cholesterol metabolism. Levrat et al. [191] studied the role of dietary propionic acid on hypocholesterolemic rats. Inulin supplementation yielded an increase in caecum SCFA compared to the control diet. However, the concentrations of SCFA measured in the portal vein indicate that the liver is exposed to high concentrations of SCFA, specifically propionic acid. The authors concluded that the presence of propionic acid in the liver reduces cholesterolemic responses. Another study reported the effect of prebiotic fiber on serum lipids and hepatic gene expression in JCR: LA-cp rats. The authors reported that prebiotic fibers caused a decrease in cholesterol levels by improved cholesterol excretion in the form of bile as well as hinder liver triacylglycerol accumulation. The authors recommended the use of prebiotic fibers as dietary treatments for hypercholesterolemia [189].

#### 4.4.3. Kidney Responses

Apart from the liver, the kidney is another essential organ affected by a significant intake of dietary fiber. Dietary fiber has been proven in numerous reports to decrease nitrogen burden as well as systemic inflammatory abuse in CKD. Similar to NAFLD, in vivo CKD experiment often depict an altered gut microbiota [192,193], intestinal inflammation, increased intestinal permeability as well as an increment in plasma concentrations of microbe- derived metabolites (e.g., p-cresol sulfate and indoxyl sulfate) [20,194,195]. It has been shown by epidemiological research that a sufficient intake of dietary fiber decreases all potential causes of mortality in patients diagnosed with CKD [196]. Although the mechanism is not fully understood, it is however attributed to the role of dietary fiber in maintaining substrate delivery to the lower gut, which in the long run modifies bacterial metabolism. Certainly, if a sufficient amount of nondigestible carbohydrates is unable to reach the colon, then substrates like amino acids will be fermented, leading to the formation of potentially harmful metabolites like p-cresol and indoles, which will consequentially stress the kidney [20,179,197].

A known mechanism highlighting the effects of dietary fiber on the kidney include the reduction in nitrogen load on both liver and kidneys via increasing microbial biomass. The microbial generated biomass aids to sequester nitrogen in the gut and decrease the amount that reaches the portal circulation [20,198]. In 2019, Adams et al. studied the effects of dietary supplementation of prebiotic fiber on nitrogen metabolism, intestinal morphometry, and blood biochemistry in weaning piglets. The inclusion resulted in an increase in nitrogen metabolism by increasing the digestibility and utilization of nitrogen as well as decreasing fecal and urinary content of nitrogen [38]. Again, Mardinoglu et al. [199] compared the metabolism of amino acids in conventional and germ-free mice, and discovered that the concentrations of amino acids entering the liver portal vein in the conventional mice were low; the reduction in the portal amino acids was associated with increased demand in nitrogen for microbial synthesis [200]. The formation of SCFAs is another proposed mechanism by which dietary fiber can impact kidney function. Thus, SCFAs are suggested to influence kidney blood flow by stimulating olfactory receptor 78 (Olfr-78), a G-protein-coupled receptor found in the renal juxtaglomerular apparatus that encourages renin secretion, which is responsible for regulating blood pressure [201]. Blood pressure regulation is an essential component in managing the progression of CKD [202]. In summary, dietary fiber is suggested to improve kidney function as well as decrease the occurrence of CKD via causing shifts in the microbiome, which will consequentially maintain and improve the gut barrier, altering the metabolism of uremic solute and microbial nitrogen, as well as controlling renal blood flow.

## 5. Conclusions

This current review collates scientific findings on the bioactive compounds in soybean residue (okara) and discusses the possible prebiotic impact of this fiber-rich residue as a functional diet on eubiosis/dysbiosis condition of the gut, as well as the consequential influence on liver and kidney functions.

The disposal of okara is still an unsolved problem. However, the functionality as well as the embedded nutritional components make it commercially suitable. The hypoglycemic effect of okara fortified foods is attributed to its dietary fiber content. Multiple reports reveal that okara possesses high nutritional value and hence can be used in the food industry to partially substitute traditional flour in order to aid increase mainly the dietary fiber fraction of food products. Dietary fiber and prebiotic impact the host by numerous mechanisms such as regulating stool bulking effects, blood glucose, or insulin levels, increasing the acidity of the gut, decreasing intestinal transit time, constructive synthesis of SCFAs, fostering the growth of beneficial gut bacteria, and hindering the growth of pathogenic ones, which in turn impact the production of microbial metabolite as well as the host immune response, and will consequently protect the kidney and liver from the translocation of pro-inflammatory bacteria. Although okara is endorsed in many contexts, it still remains vital to adopt an appropriate technique such as enzymatic and chemical treatment, fermentation, high pressure, extrusion, micronization, and further grinding into powdered form for its effective utilization and acceptance. In addition, further exploration and experimentation are needed to provide an in-depth understanding of the nutraceutical components as well as functional foods fortified with okara to aid confirm the specific microbes and signals that are transformed in response to this dietary fiber.

## Figures and Tables

**Figure 1 molecules-26-00326-f001:**
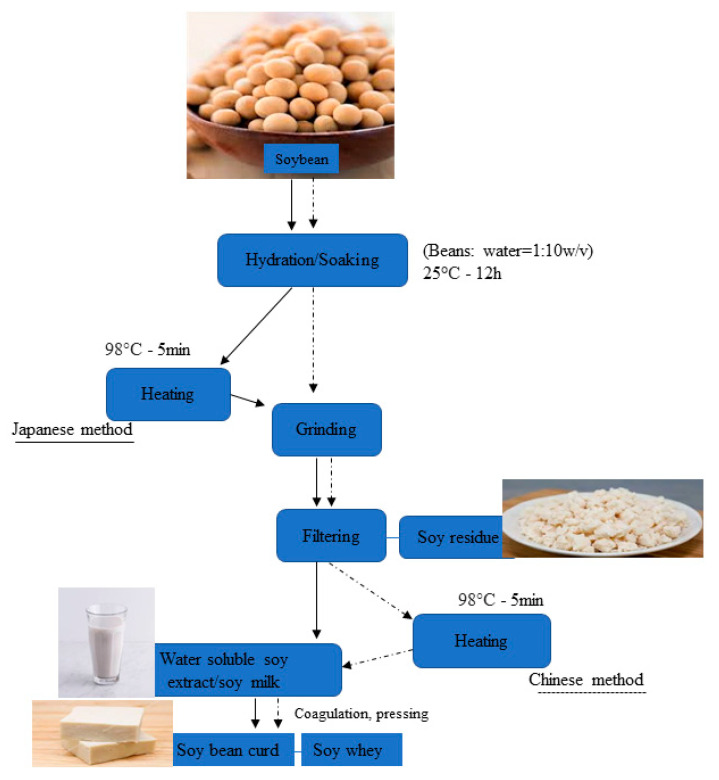
Schematic illustration of the steps involved in processing soymilk and the production of soybean residue/okara.

**Figure 2 molecules-26-00326-f002:**
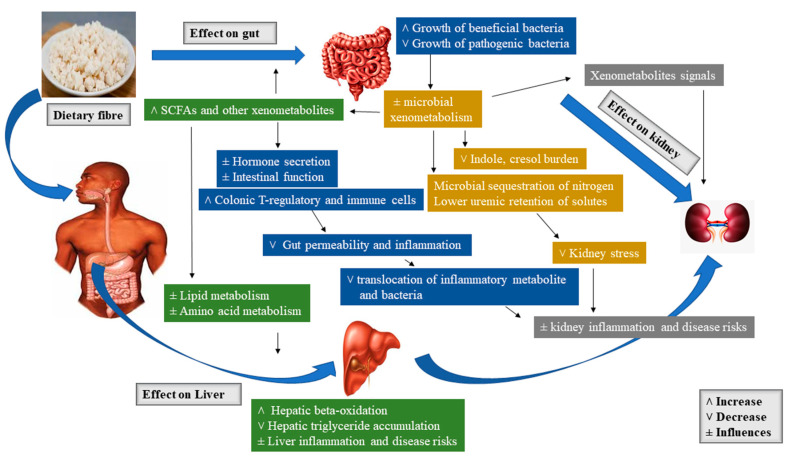
Schematic overview of the major prebiotic effects of dietary fiber on gut, liver, and kidney.

**Table 1 molecules-26-00326-t001:** Effect of heat, fungi, and bacterial treatments on soy and soy-based products.

Soy Products	Treatments (Bacterial Fermentation)	Health Benefits	References
	**Starter culture**		
Fermented soybean	*Bacillus subtilis* SHZ, *B. subtilis* MTCC 5480	Antioxidant	[62,63]
Douchi qu	B. subtilis natto, *B. subtilis* B1	ACE inhibitory	[64]
Cheonggukjang	*B. licheniformis* SCD 111067P	Antihypertensive, Antidiabetic	[65,66]
**Soy Products**	**Treatments (Fungi Fermentation)**	**Health Benefits**	**References**
	**Starter culture**		
Douchi qu	*Aspergillus oryzae*, *Mucor wutungkiao*	ACE inhibitory	[64]
Douchi	*Aspergillus oryzae*, Aspergillus *egyptiacus*	Antioxidant, Antihypertensive	[67,68]
Meju	Aspergillus oryzae	Antimicrobial	[69]
**Soy Products**	**Heat Treatments**	**Effects**	**References**
Soymilk	Microwave-assisted extraction	Increase protein content, viscosity, protein solubility, and digestibility	[70]
Raw soybean	Gamma irradiation	Increase total phenolic content, Decrease tannins and trypsin inhibitors	[71]
Soybeans	Infrared treatment	Inactivated both trypsin inhibitors and lipoxygenase	[72]

ACE—Angiotensin-converting enzyme.

**Table 2 molecules-26-00326-t002:** General nutritional constituents of soybean residue (okara).

Micro Components	Amount (mg/100 g Dry Matter)
Thiamine (B1)	0.48–0.59
Niacin (B3)	0.82–1.04
Riboflavin (B2)	0.03–0.04
Mn	0.2–3.1
Zn	0.3–3.5
Cu	0.1–1.2
Fe	0.6–11
Na	16–96
Ca	260–428
K	936–1350
Mg	130–165
**Macro Components**	**Amount (g/100 g Dry Matter)**
Ash	3.0–4.5
Dietary fiber	42.4–58.1
Soluble dietary fiber	4.2–14.6
Insoluble dietary fiber	40.2–50.8
Fat	8.3–10.9
Carbohydrate	3.8–5.3
Protein	15.2–33.4
**Phytochemicals**	**Amount (g/100 g Dry Matter)**
Isoflavone glucosides	10.3
Isoflavone aglycones	5.41
Malonyl glucosides	19.7
Acetyl glucosides	0.32
Saponins	0.1
Phytic acid	0.5–1.2

Adapted from Vong and Liu [46].

**Table 3 molecules-26-00326-t003:** Effect of okara supplementation on nutrition (in vivo/in vitro).

Experimental Model	Effect	Conclusion Remarks	Reference
In vivo (Wistar rats fed high-fat diet), fed okara (20%), for four weeks.	↓ body weight, ↓ triglycerides, ↑ SCFA production, ↑ amino acid metabolism, ↑ mineral absorption, ↑ microbial protection; ↔ *Firmicutes*: *Bacteroidetes* ratio, *Bacteroides* and *B*. *coccoides-E*. *rectale* groups in control group, ↑ *C. leptum* and *Bacteroides* population in feces, *Enterobacteriaceae* (cecal content) and *Enterococcus* (fecal and cecal content) groups.	Okara exerts health-promoting attributes in vivo and could be further used as prebiotic and functional ingredient in foods	[51]
In vivo (Female Wistar rats fed a standard rat diet) fed dietary rich okara (10%), for four weeks	↓ body weight gain, ↓ total cholesterol, ↑ antioxidant status and butyrogenic effect in the cecum, ↑ apparent absorption and true retention of calcium	The development of an innovative soybean by-product rich in dietary fiber could be useful as a functional ingredient with health-promoting benefits	[8]
In vivo (High-cholesterol-fed Wistar rats), for four weeks	↓ liver and serum triglyceride levels, ↓ pH of fecal contents, ↑ total lipids, triglycerides and bile acids in feces, ↑ SCFA production.	Enzymatically treated okara fiber can improve intestinal transit by increasing fecal bulk.	[6]
In vitro (Water jet (WJ) treated okara and water jet treated microcrystalline cellulose (MCC)), effect on α-amylase inhibition and butyrate production using *Roseburia intestinalis*	↑ inhibition of α-amylase activities by WJ-treated okara than WJ-treated MCC, ↑ butyrate production by *Roseburia intestinalis* in WJ-treated okara	These results depict that WJ system can be used on okara to improve inhibited α-amylase activities and butyrate production by gut microbiota.	[125]
In vivo (High fat-fed Syrian hamsters), fed 13% or 20% okara fiber for 3 weeks	↑ fecal excretion of total lipids, triglycerides, free cholesterol, and total nitrogen. ↔ feed intake and body weight gain. 20% okara group. ↓ plasma triglycerides, VLDL- plus LDL cholesterol and total cholesterol, ↓ liver total lipids, triglycerides, and total esterified cholesterol concentrations.	Okara might aid in the prevention of hyperlipidemia and could be used as a natural ingredient for functional food preparation.	[12]
In vitro (Fermentability and prebiotic potential of okara using human fecal slurries), using 16S rRNA-based fluorescence in situ hybridization, and HPLC	↑ SCFA plus lactic acid, ↑ beneficial bacteria (*bifidobacteria*l and *lactobacilli*), ↓ potentially harmful bacterial groups (*clostridia* and *Bacteroides*)	The differences observed between fructo-oligosaccharides and okara substrates could be accredited to the great complexity of okara’s cell wall, which requires longer times to be fermented than other easily digested molecules. Hence, allowing an extended potential prebiotic effect. These findings support an in vitro potential prebiotic effect of Okara.	[126]
In vivo (Wistar Hannover female rats), control group (fed standard rat chow) and treated group (fed a mixture of the standard rat chow plus okara), for 4 weeks	↔ food intake, ↓ growth rate, and feeding efficiency, ↑ fecal weight and moisture, ↓ lower pH, ↑ cecal weight, ↑ total SCFA production in okara-fed group than control group. ↔ albumin, uric acid, protein, bilirubin, or glucose content in rat serum for both groups.	Okara is a rich source of low-cost dietary fiber and protein, and might be effective as a dietary weight-loss supplement with a potential prebiotic effect.	[127]
In vivo (Senescence-accelerated mouse prone 8 (SAMP8) mice), fed standard diet, or a diet containing (7.5% or 15%, *w*/*w*) okara, for 26 weeks	15% okara-fed group; ↓ body weight, ↑ fecal weight, and altered cecal microbiota composition compared with the control group, ↔ serum lactic acid, and butyric acid levels. 7.5% okara-fed group; ↑ NeuN intensity in the hippocampus than control mice, ↓ inflammatory cytokine TNF-α, ↑ brain-derived neurotrophic factor, ↑acetylcholine synthesizing enzyme ↑ acetylcholine level in the brain	Oral administration of okara could delay cognitive decline without drastically changing gut microbiota	[128]
In vivo (high fat-fed C57BL/6J male mice), for 12 weeks	↓ body weight and epididymal fat weight. ↓ serum and hepatic lipid profiles. ↑ fecal triacylglycerol and total cholesterol levels. ↑ PPAR-α expression, ↓ PPAR-γ and FAS levels	Okara consumption appears to protect mice against diet-induced obesity and metabolic dysregulation related to obesity	[129]
In vivo (high lard fed Goto-Kakizaki (GK) Type 2 diabetes Male rats), for 2 weeks	↔ body weight gain or food intake, ↓ plasma glucose levels, ↑ mRNA expression levels of PPARγ, adiponectin, and GLUT4,	The study suggested that okara can play a significant role in treating type 2 diabetes.	[130]
In vivo (Human Type 2 Diabetes Mellitus outpatients), fed okara for 2 weeks	↑ food intake (fiber 6.9 to 12.6 g), ↓ Fasting blood glucose (6.3 to 5.4 mmol/L), ↓ fructosamine (319 to 301 μmol/L)	Okara increased fiber intake and consequently improved blood glucose in DM patients	[131]

**Table 4 molecules-26-00326-t004:** Impact of okara fortification on food properties and function.

Experimental Model	Dietary Formulations	Effect on Food Properties and Function	Description	Reference
In vitro (Okara and vital gluten on physicochemical properties of noodle).	Added portion of okara (0%, 5% & 10%)	↑ total phenolics, flavonoids and radical-scavenging activity. 10–15% okara ↓ optimum cooking time, extensibility tensile strength, and elasticity of noodle.	5% or 10% dried okara powder plus 6% vital gluten might be best in producing noodles with increased phytochemicals and consumer’s sensory acceptability.	[106]
In vitro (Application of okara to enrich vegetable paste)	High moisture (80.77–81.42%). Low lipid (5.62–7.62%). Low calorie (95.14–108.14 kcal).	↑ β-carotene (0.411 mg/100 mL). ↑ antioxidant activity. ↑ isoflavones (0.15 μmol/gFM).	The sample with the lowest content of okara (34 g/100 g) presented the highest average of 8.0 in the acceptance test and was also considered the tasters’ favorite one.	[132]
In vitro (Starch digestibility of steamed rice bread fortify with okara)	Added portion of okara (0%, 7%, 14% & 21%)	↑ Elasticity and viscidity. ↓ Hardness, cohesiveness, and chewiness. ↑ Amylose content, slowly digestible starch, resistant starch. ↓ Predicted glycemic index (pGI) from 79.14 to 74.17–68.91	Okara can potentially modify the texture and starch digestibility of steamed rice bread.	[133]
In vitro (Gastrointestinal stress in synbiotic soy yogurt with okara during storage), for 28 days	Soy yogurt + Okara, Soy yogurt + Guava pulp Soy yogurt + Mango pulp	↑ Survival rates (%) of *L. acidophilus* La-5 and *B. animalis* Bb-12, ranging from 8 to 9 log cfu/g after simulated gastrointestinal conditions. ↑ Probiotic strains functionality.	In this study, okara endorsed probiotic functionality in simulated gastrointestinal conditions, however, the addition of fruit pulps might lead to a reduction.	[134]
In vitro (Digestibility of rice noodle enriched with okara)	Added portion of okara (0%, 5%, 10%, & 20%).	↑ cooking loss, adhesiveness, and hardness with increasing level of okara. ↓ water absorption, cohesiveness, and swelling index. 0%, 5% & 10% okara ↓ in vitro starch digestibility. 10% okara ↓ predicted glycemic index	10% okara can be used to produce health-beneficial rice noodles with reduced in vitro starch digestibility and improve cooking quality.	. [15]
In vitro (Digestibility and structural attributes of okara-enriched functional pasta)	Added okara contents (10–50%)	↔ structural changes, ↓ glycemic index (27.41 ± 0.05–12.38 ± 0.01). 50% okara ↑ total phenolic content and antioxidant activity (158.37 ± 0.40 to 232.90 ± 0.85 mg GAE/100 g and 10.87 ± 0.10%–56.21 ± 0.05%)	The study indicated that pasta enriched with okara has the potential to be commercialized on the industrial level to develop nutritional enriched functional pasta.	[107]

Note: The symbols denote a significant increase (↑), decrease (↓), and (↔) no significant difference observed after okara application.

## Data Availability

No available data. All relevant and required data were extracted from the included articles and are all duly cited, and/or can contact the corresponding authors or the first author for any further clarifications.

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
