# Peer review of "Prebiotic Impacts of Soybean Residue (Okara) on Eubiosis/Dysbiosis Condition of the Gut and the Possible Effects on Liver and Kidney Functions"

_molecules, 2021, doi:10.3390/molecules26020326_

Round 1
Reviewer 1 Report
Manuscript ID: molecules-1021919
Title: Prebiotic Impacts of Soybean Residue (Okara) on Eubiosis/Dysbiosis Condition of The Gut and The Possible Effects on Liver and Kidney Functions
Major comments:
Abstract did not showed novelty of this review, therefore need to revise.
Introduction section very poor written and background literature information are missing, therefore need to revise with additional information to support this review. Also, research gap needs to highlight in the introduction section.
Figure 2 quality is very poor, need replace with improved figure with better quality.
The extensive review needs to have comprehensive tables and this review have only one table with General nutritional constituents of soybean residue. Therefore, please add at least 4-5 table to highlight the application of soybean residue (okara) in different sectors.
Conclusion section need to be more concise, therefore need significant revision.
Author Response
Thank you for the guidance and corrections, and making the manuscript better.
The manuscript has been carefully revised following the comments given.
- Abstract: The abstract has been revised completely and can be seen on line 14-31 on the manuscript.
- Introduction: The introduction has as well been revised starting from line 43-81 of the manuscript
- Figure 2: The figure has been replaced with an improved and high quality (300dpi) version. It can been seen on line 637 of the revised manuscripts.
- Comprehensive tables: The manuscript is updated with additional 3 tables making a total of 4. Named; Table 1. Effect of heat, fungi, and bacterial treatments on soy and soy-based products. (line 199),Table 2. General nutritional constituents of soybean residue (okara) (line 332),Table 3. Effect of okara supplementation on nutrition (in vivo/in vitro) (line 453).Table 4. Impact of okara fortification on food properties and function (line 455)
- Conclusion: The conclusion has been summarised and updated also. can be seen on line (759-799).
Reviewer 2 Report
This work is interesting and well detailed, however, the following comments should be taken into account so that its publication in Molecules can be accepted.
The use of the appropriate terms of valorization, residues by-products, and co-products are not the most appropriate. Please use the most appropriate word for each case. Is not the same as the use of a residue or a co-product in foods.
Line 89-90 this sentence is inaccurate since it should be rewritten if the meaning of it is wanted to be preserved since there is the dietary fiber of "animal origin".
Line 93-94, another inaccurate phrase, should be rewritten and use the more appropriate word, animal-based foods.
In my opinion, not only the health properties of fiber should be specified, but also their physicochemical role in the gastrointestinal tract, key for these beneficial effects to take place as well as their interaction with the gut microbiota. This aspect must be included
Line 197, it is more correct to speak of techno-functional properties than of functional properties
line 238, please do not use the word residue, use co-products
Author Response
Thank you for the guidance and corrections, and making the manuscript better.
- Use of the appropriate terms: The entire manuscript have been revised and all soybean residue and/or by-product have been replaced with a single word (Okara) across the manuscript.
- Line 89-90 of old manuscript have been revised and new definition for fibre has been replaced (line 92) of the revised manuscript.
- line Line 93-94 of old manuscript have been deleted.
- As advised, new subtopic have been created on the physicochemical role of dietary fibre and can be found on line (481) 4.1 Physicochemical Role of Dietary Fibre in The Gastrointestinal Tract
- techno-functional properties (Line 197) of old manuscript is revised and update in new manuscript can be seen on line(203-206)
- Residue in (line 238) of old manuscript has been replaced with co-products, as advised and can be seen on (line 246) of the revised manuscript.
Reviewer 3 Report
This review article describes the functional properties of soybean residues derived from the grinded soy, focusing on the prebiotic effects of dietary fibers included in the soybean residue. Soy contains diverse bioactive components, and the soybean residue is rich in phytochemicals and dietary fibers. Considering the role of foods, intestine is a central organ to absorb the nutrients included in foods. Furthermore, intestinal function is associated with gut microbiota. Therefore, the soybean residue is precious source of dietary fiber and the valorization of soybean residue is important for not only reducing the food loss but also helping keep our health.
The topic which this article deals with is the latest and valuable from the viewpoint of food science and medicine. Section 2 is very interesting and easy-to-understand in order to know the details of soybean residue as food component. Section 3 is also a good review for the functional benefits of soybean residue which is exerted via restoring eubiosis of gut microbiota and improving the intestinal environment, resulting in ameliorating the disorders of gut, liver and kidney. Section 4, however, is rather the summary of prebiotics but not of soybean residue. Therefore, the reader will be slightly confused in considering whether the effect on gut, liver and kidney is specifically regulated by soybean residue-derived fiber or other food-derived fiber.
The topic is up-to-date and the reference is enough. I recommend that the context of Section 4 will be revised so as the readers to know that the evidence referred is limited to soybean-derived fiber or generally expanded to other food-derived fiber. If the reader will understand the difference between soy and other foods, the importance of this review will become no doubt much higher.
Author Response
Thank you for the guidance, corrections, and advise in making the manuscript better.
Your comment was right. The section 4 is a summarised on general fibre.
Hence as advised, the main heading for section 4, has been updated and dietary fibre is now included, together with other two subheadings 4.1(newly added subheading) and 4.4
4.0 Physicochemical and Prebiotic Influence of Dietary Fibre (Emphasis on Okara-Derived Fibre) on Gut and Associated Tissues: Gut, Liver, and Kidneys (line 481) of the revised manuscript.
4.1 Physicochemical Role of Dietary Fibre in The Gastrointestinal Tract (line 481) of the revised manuscript.
4.4 Gut, Liver, and Kidney responses to the prebiotic effects of dietary fibre in General (line 624) of the revised manuscript.
Round 2
Reviewer 1 Report
Authors improved the manuscript.